

# High-dose testosterone supplementation disturbs liver pro-oxidant/antioxidant balance and function in adolescent male Wistar rats undergoing moderate-intensity endurance training

Ewa Sadowska-Krępa[1], Barbara Kłapcińska[1], Anna Nowara[1], Sławomir Jagsz[1], Izabela Szołtysek-Bołdys[2], Małgorzata Chalimoniuk[3], Józef Langfort[1] and Stanisław J. Chrapusta[4]

[1] Institute of Sport Sciences, The Jerzy Kukuczka Academy of Physical Education, Katowice, Silesian Voivodeship, Poland
[2] Department of General and Inorganic Chemistry, School of Pharmacy with the Division of Laboratory Medicine in Sosnowiec, Medical University of Silesia, Sosnowiec, Silesian Voivodeship, Poland
[3] Department of Physical Education and Health in Biała Podlaska, Józef Piłsudski University of Physical Education in Warsaw, Biała Podlaska, Lublin Voivodeship, Poland
[4] Department of Experimental Pharmacology, Mossakowski Medical Research Centre, Polish Academy of Sciences, Warsaw, Mazowieckie Voivodeship, Poland

Corresponding author
Stanisław J. Chrapusta,
sjchrapusta@imdik.pan.pl

## ABSTRACT

In some countries, anabolic-androgenic steroid abuse is rampant among adolescent boys and young men, including some of those seeking physical fitness and/or pleasing appearance through various exercise types. This tactic carries the risk of severe harmful health effects, including liver injury. Most anabolic-androgenic steroid stacking protocols employed are based on the use of the 'prototypic' anabolic-androgenic steroid testosterone and/or its esters. There is a vast body of data on the effects of anabolic-androgenic steroids' abuse combined with physical exercise training on the liver antioxidant barrier in adult subjects, whereas those concerning adolescents are scant. This study aimed to assess, in adolescent male Wistar rats undergoing a 6-week moderate-intensity endurance training (treadmill running), the influence of concurrent weekly supplementation with intramuscular testosterone enanthate (TE, 8 or 80 mg/kg body weight/week) on selected indices of liver status and oxidative stress. The rats were sacrificed, and their livers and blood samples were harvested two days after the last training session. High-dose TE treatment significantly reduced body and liver weight gains. Neither low-dose nor high-dose TE treatment affected liver $\alpha$-tocopherol or $\gamma$-tocopherol content, whereas low-dose TE treatment significantly lowered hepatic reduced glutathione content. TE treatment significantly elevated liver thiobarbituric acid-reactive substances content and blood activities of alkaline phosphatase and $\gamma$-glutamyltransferase, but not of aspartate aminotransferase or alanine aminotransferase. Liver catalase activity was lowered by >50% in both TE-treated groups, while superoxide dismutase activity was significantly but slightly affected ($-15\%$) only by the high-dose TE treatment. Glutathione peroxidase and glutathione reductase activities were not significantly altered. TE treatment significantly
increased liver thiobarbituric acid-reactive substances content and lowered blood HDL-cholesterol, but did not significantly affect LDL-cholesterol or triglycerides level. In conclusion, high-dose TE treatment significantly disturbed liver antioxidant barrier and prooxidative-antioxidative balance and hence counteracted favorable effects of concurrent moderate-intensity endurance training in adolescent male rats.

# INTRODUCTION

Endurance training (EndTr) plays a key role in various endurance sports, e.g., cross-country skiing, long-distance running, and cycling. However, it can also be a vital addition to both bodybuilding (M. Barroso. 8 tips for balancing bodybuilding and endurance training; for the URL, see Supplemental File) and strength training, particularly in team sports (*Coffey & Hawley, 2017*). It also gets growing attention from the general public as a health-promoting factor when used in moderation (*Fikenzer et al., 2018*; *Ruegsegger & Booth, 2018*); the positive effects of the training also concern liver health (*Shephard & Johnson, 2015*).

Some adolescents and young adults, mostly males (*Kanayama & Pope Jr, 2018*; *Johnston et al., 2019*), combine endurance-oriented physical training with the use of testosterone and/or its synthetic derivatives termed anabolic-androgenic steroids (AAS). The main goal of this tactic is a faster reduction of body fat and boosting muscularity with the idea of improving physical performance and appearance (*Hartgens & Kuipers, 2004*; *Kanayama & Pope Jr, 2018*), and hence self-esteem. A positive correlation of the bodily effects with dosage inspires AAS abuse at high doses. Some endurance athletes reported the effectiveness of AAS for accelerating the recovery after intense physical exercises (*Hartgens & Kuipers, 2004*). AAS were also reported to improve running endurance in male rats (*Van Zyl, Noakes & Lambert, 1995*; *Georgieva & Boyadjiev, 2004*), but no positive AAS effect on endurance or blood serum markers recovery was seen in other rat studies (*Delgado, Saborido & Megias, 2010*) or healthy men (*Baume et al., 2006*).

Xenobiotics and excess endobiotics are mostly processed and removed by the liver; this situation renders this organ the critical site of steroid toxicity (*Russmann, Kullak-Ublick & Grattagliano, 2009*; *Vinken et al., 2013*). The abuse of AAS and their 'prototypic' testosterone is associated with a variety of severe adverse health effects (*Hartgens & Kuipers, 2004*; *Van Amsterdam, Opperhuizen & Hartgens, 2010*; *Vanberg & Atar, 2010*), including a number of those directly linked to liver injury (*Russmann, Kullak-Ublick & Grattagliano, 2009*; *Bond, Llewellyn & Van Mol, 2016*; *Solimini et al., 2017*). A growing body of data links this damage to a variety of genomic and nongenomic actions of these drugs, including enhanced generation of reactive oxygen species and lipid peroxidation, and the related activation of cellular stress-signaling pathways (*Russmann, Kullak-Ublick & Grattagliano, 2009*; *Cerretani et al., 2013*; *Bond, Llewellyn & Van Mol, 2016*). Disruption of redox homeostasis is a well-established event in both drug hepatotoxicity and various

liver diseases (*Cichoż-Lach & Michalak, 2014*; *Li et al., 2015*; *Arauz, Ramos-Tovar & Muriel, 2016*). The primary sources of reactive oxygen species, the noxious mediators of oxidative stress, are cytochrome P450 enzymes of the endoplasmic reticulum and mitochondria (*Guengerich, 2008*; *Jones, 2008*) and several enzymes present in peroxisomes that abound in the liver (*Cerretani et al., 2013*; *Cichoż-Lach & Michalak, 2014*).

Hepatocytes carry several systems capable of preventing or limiting the adverse effects of enhanced oxidative stress (*Jones, 2008*; *Frankenfeld et al., 2014*; *Sies, 2015*). These systems include various antioxidant enzymes, e.g., glutathione peroxidase (GPx), glutathione reductase (GR), superoxide dismutase (SOD) and catalase (CAT), and low molecular weight antioxidants, e.g., tocopherols and reduced glutathione (GSH) that act mostly in lipophilic and hydrophilic milieus, respectively. Data on the effects of AAS abuse combined with physical exercise training on the liver antioxidant barrier in adolescent subjects are scarce (*Molano et al., 1999*; *Pey et al., 2003*). Adolescence is a period when some people, mainly boys, become familiar with AAS as drugs of abuse. It is also a period of essential changes in androgen catabolism in men (*Horst, Bartsch & Dirksen-Thiedens, 1977*; *Belgorosky & Rivarola, 1987a*; *Belgorosky & Rivarola, 1987b*; *Stárka, Pospíšilová & Hill, 2009*), which condition may significantly modify the outcome of androgen action (*Mantovani & Fucic, 2014*).

Our earlier study on the effects of long-term testosterone treatment on the liver antioxidant barrier and some blood markers of liver injury in sedentary adolescent male rats showed some signs of enhanced liver oxidative stress and toxicity but no potential lasting harm (*Sadowska-Krępa et al., 2017*). Here, we tested the possible harmful effects of this treatment in a situation aimed to model androgen abuse to aid physical exercise training. The studied constituents of hepatic antioxidant defense systems included selected antioxidant enzymes (SOD, CAT, GPx, and GR) and nonenzymatic low molecular weight antioxidants ($\alpha$- and $\gamma$-tocopherols and GSH); liver thiobarbituric acid-reactive substances (TBARS) content was assayed as an index of oxidative stress. Our choice of testosterone enanthate (TE) was mainly based on the fact that testosterone was for many years among the most frequently abused doping drugs (*Handelsman, 2006*), and testosterone formulations (chiefly esters) are a cornerstone of most oral and injectable AAS stacking regimens (for the respective URLs see Supplemental File). However, we also intended to avoid hepatotoxicity related to chemical alteration (mostly $17\alpha$-alkylation) of the sterane core in most AAS (*Hartgens & Kuipers, 2004*; *Russmann, Kullak-Ublick & Grattagliano, 2009*; *Büttner & Thieme, 2010*).

## MATERIALS & METHODS

### Animals

Five-week-old healthy specific pathogen-free male outbred Wistar rats of 91–120 g initial body weight (BW) from the Cmd:(WI)WU stock maintained at the Mossakowski Medical Research Centre, Warsaw, Poland, were used for the study. They were housed 4–5 per opaque plastic cage (60 × 38 cm floor size) with dust-free deciduous wood chip bedding, in a controlled environment room (22–24 °C, 45–65% relative humidity, 15–20 air

changes per hour), under 12h/12h light-dark cycle (lights on at 7 a.m.). Throughout the study (excepting an 8-hour fast just before the sacrifice, see below), the rats were allowed autoclaved laboratory rat maintenance chow (ssniff Spezialdiäten GmbH, Soest, Germany) and autoclaved purified tap water *ad libitum*. The cage environment was enriched with deciduous wood shavings as a material for gnawing and nesting; after each bedding change, a few fresh chow pellets were also left on the cage floor for playing and gnawing. The water bottles were replaced twice a week, and the bedding and enrichment material were changed weekly or more frequently if needed.

## Drugs

Stock TE solution (*Testosteronum prolongatum*, Jelfa, Jelenia Góra, Poland; active substance: testosterone enanthate, 100 mg/ml) was diluted with sesame oil (Sigma-Aldrich, St. Louis, MO, USA) as necessary to provide the same injection volume of 1 ml/kg$_{BW}$ irrespective of weekly TE dose employed. The diluted solution was injected intramuscularly each Monday for six weeks, alternatively into the left and right hind leg. TE-untreated rats received 1 ml/kg$_{BW}$ of the oil by an identical schedule.

## Experimental design

All rats meant for EndTr were first run-tested (at 18–20 m/min, 0° slope, 3 × 5 min daily, with 15-min breaks) for three consecutive days on a BTP-10 motorized rodent treadmill (Porfex, Białystok, Poland) to acquaint them with this setting and identify and exclude rats reluctant to exercise. A low-intensity electrical shock (0.5 mA, 170 V AC) was used during the habituation to motivate the rats to run. Two rats that were found unwilling to run during the habituation period did not enter the experiment. The remaining rats were randomly divided between three groups: (1) TE-untreated EndTr rats (EndTr, $N = 11$), (2) EndTr rats given 8 mg/kg$_{BW}$/week of TE (EndTr+TE8, $N = 11$) and (3) EndTr rats given 80 mg/kg$_{BW}$/week of TE (EndTr+TE80, $N = 12$). An additional group of naïve sedentary (untrained, UTr, $N = 11$) male Wistar rats from the same stock and kept under the same conditions, 11–12-week old at the moment of sacrifice, was used to estimate reference ranges of blood serum lipids. All the rats belonging in the EndTr, EndTr+TE8, and EndTr+80 group were trained to run (at 0° slope) on the treadmill five days a week (Monday through Friday) for six weeks starting two days after the last habituation session. Treadmill speed was gradually elevated from 16 m/min for the first week to 28 m/min for the fourth week and then was kept steady. The EndTr session duration for weeks 1–4 began at 40 min/day each week and was extended by 5 min daily; for the last two weeks, the rats ran for 60 min daily. This moderate-intensity EndTr (at about 60% VO$_2$max) was shown by various measures to improve endurance in rats (*Langfort, Budohoski & Newsholme, 1988*; *Langfort et al., 1996*; *Dobrzyn et al., 2013*). One rat of the EndTr group and two rats of the EndTr+TE8 group developed an aversion to run and showed noticeable weight loss (by ≥10% over a single week) at some time point of the training course. These rats were euthanized by decapitation while deeply anesthetized with an intraperitoneal injection of a solution of pentobarbital sodium and pentobarbital (50 mg/ml and 10 mg/ml, respectively; Vetbutal, Biowet Puławy, Poland) at a dose of 80 mg/kg$_{BW}$. Two days

after the last EndTr session, all the surviving endurance-trained rats and the UTr rats were fasted for 8 h, anesthetized with an intraperitoneal injection of the pentobarbital sodium and pentobarbital mixture as above, and decapitated. Trunk blood samples were collected, let to clot at room temperature, and centrifuged to yield serum for biochemical assays. Livers were perfused *in situ* with 10 mM glucose-supplemented cold Krebs-Henseleit buffer pH 7.4, then quickly removed, weighed, and cut into several pieces that were instantly frozen in liquid nitrogen and stored at −80 °C until analyzed. The study protocol complied with the Directive 2010/63/EU of the European Parliament and of the Council of 22 September 2010 on the protection of animals used for scientific purposes, was in line with the respective Polish law then in force, and was accepted by the IV Local Ethics Committee for Animal Experimentation in Warsaw, Poland (Permit No. 38/2011).

## Assay methods

All biochemical determinations in blood serum and liver samples, except for serum lipids that were not determined before, were performed as described earlier (*Sadowska-Krępa et al., 2017*). Specifically: The total serum testosterone (TT) level was assessed with a DSL-4100 Testosterone RIA Kit (Diagnostic Systems Laboratories, Webster, TX, USA). Serum activities of aspartate aminotransferase (AST; EC:2.6.1.1), alanine aminotransferase (ALT; EC:2.6.1.2), alkaline phosphatase (ALP; EC:3.1.3.1), and $\gamma$-glutamyltransferase (GGT; EC:2.3.4.2) were determined on a model Cobas Integra 400/800 analyzer (Roche, Switzerland). For antioxidant status testing, liver samples were homogenized in ice-cold buffers prepared according to the respective diagnostic kit instructions, using a model Ultra-Turrax T8 homogenizer (IKA Labortechnik, Staufen, Germany). Protein content in homogenate supernatants was determined with the BCA-1 Protein Assay Kit (Sigma-Aldrich, UK). CAT (EC:1.11.1.6) activity was measured as described by *Aebi (1984)*, while GSH content and GPx (EC:1.11.1.9) and GR (EC:1.6.4.2) activities were assessed with Bioxytech kits GSH-400, GPx-340, and GR-340, respectively (OXIS International, Portland, OR, USA). SOD (EC:1.15.1.1) activity was assessed with a Superoxide Dismutase Assay Kit (Cayman Chemical, Ann Arbor, MI, USA). Liver contents of $\alpha$- and $\gamma$-tocopherol were quantified by HPLC (*Sobczak, Skop & Kula, 1999*), while that of thiobarbituric acid-reactive substances (TBARS) was assessed as described by *Ohkawa, Ohishi & Yagi (1979)* and expressed in malondialdehyde (MDA) units. Serum total cholesterol (Tot-Ch), HDL-cholesterol (HDL-Ch), and triglyceride (TG) levels were determined using commercial kits (CH-200, CH-20, and TR-210, respectively) from Randox Laboratories (Crumlin, UK), while LDL-cholesterol (LDL-Ch) was assessed with a model Synchron CX9 Pro analyzer (Beckman-Coulter).

## Statistics

Data are presented as the mean ± SD if applicable. Body weight data were first analyzed by a two-way analysis of variance (ANOVA) with weekly TE dose (0, 8, or 80 mg/kg$_{BW}$) as the main factor and repeated measure on time, followed by the Tukey test for unequal sample sizes. Blood serum lipid titers were compared by a one-way ANOVA followed by the Dunnett test. The occurrence of abnormal values of blood serum enzyme activities and
**Table 1 Comparison of body weight, liver weight and blood testosterone level between adolescent male rats given 6-week EndTr without or with concurrent weekly testosterone enanthate treatment.**

| Variable | Rat group | | | ANOVA results |
|---|---|---|---|---|
| | EndTr | EndTr+TE8 | EndTr+TE80 | |
| Initial BW [g] | $102 \pm 5$ | $111 \pm 7$ | $109 \pm 7$ | TE dose: $F_{2,28} = 19.2, p < 10^{-3}$ |
| | (10) | (9) | (12) | Time: $F_{1,28} = 2165.4, p < 10^{-3}$ |
| Final BW [g] | $301 \pm 27$ | $310 \pm 12$ | $252 \pm 21^{***,\#\#\#}$ | TE dose $\times$ time interaction: |
| | (10) | (9) | (12) | $F_{2,28} = 25.9, p < 10^{-3}$ |
| LW [g] | $10.30 \pm 1.41$ | $12.40 \pm 1.59^{*}$ | $8.19 \pm 1.39^{**,\#\#\#}$ | $F_{2,28} = 21.7, p < 10^{-3}$ |
| | (10) | (9) | (12) | |
| LW/BW [%] | $3.42 \pm 0.33$ | $3.99 \pm 0.45^{*}$ | $3.25 \pm 0.48^{\#\#}$ | $F_{2,28} = 8.05, p = 0.0017$ |
| | (10) | (9) | (12) | |
| TT [nmol/l] | $3.24 \pm 2.07$ | $5.02 \pm 1.59$ | $34.17 \pm 5.83^{***,\#\#\#}$ | $F_{2,28} = 181.9, p < 10^{-3}$ |
| | (7) | (9) | (11) | |

**Notes.**

BW, body weight; LW, liver weight; EndTr, endurance training; TE8, 8 mg/kgBW/week of intramuscular testosterone enanthate; TE80, 80 mg/kgBW/week of intramuscular testosterone enanthate; TT, total blood serum testosterone. Data are mean $\pm$ standard deviation; rat numbers are shown in parentheses. * $p < 0.05$, ** $p < 0.01$, *** $p < 0.001$ vs. the respective EndTr group value; ## $p < 0.01$, ### $p < 0.001$ vs. the respective EndTr+TE8 group value; Tukey's test.

lipid profile indices was compared by the one-tailed $z$-test for two proportions. All other data were analyzed by a one-way ANOVA with weekly TE dose as the main factor, followed by the Tukey test when appropriate. Comparisons with data from our previous study (*Sadowska-Krępa et al., 2017*) were made using Student's $t$-test for independent variables, as indicated in the text. Associations between variables were assessed using the Spearman rank correlation test. In all cases, $p \leq 0.05$ was considered significant. All the statistical analyses were run using the Statistica v. 12.5 software package (StatSoft, Tulsa, OK, USA).

# RESULTS

## Serum total testosterone level and body and liver weights

At the end of the study, the mean serum TT level in low-dose TE-treated (EndTr+TE8) rats exceeded only marginally and nonsignificantly that in the TE-untreated rats, while that in their high TE dose-treated counterparts (EndTr+TE80) was 10-fold higher. There was no difference in BW gain between the rats given no or low-dose TE treatment, whereas both these groups showed considerably higher BW gain than the EndTr+TE80 group. Mean liver weight and LW/BW ratio in the EndTr+TE8 rats were significantly higher, while the mean LW but not mean LW/BW ratio was significantly lower in the EndTr+TE80 rats than in the TE-untreated rats. The final BW and the LW/BW ratio were significantly lower in the high-dose- than in the low-dose TE-treated rats (Table 1). Across all three EndTr groups combined, the final BW, LW, and LW/BW ratio correlated negatively with the weekly TE dose: $R_S = -0.65, p < 0.001$, $R_S = -0.47, p < 0.01$, and $R_S = -0.37, p < 0.05$, respectively, $N = 31$ for all.

**Table 2** Liver levels of selected antioxidant enzymes, non-enzymatic antioxidants and TBARs in adolescent male rats given 6-week endurance training without or with concurrent testosterone enanthate treatment.

| Antioxidant enzyme activity, or low molecular weight antioxidant or TBARS content[b] | Rat group | | | ANOVA results |
|---|---|---|---|---|
| | EndTr N = 10 | EndTr+TE8 N = 9 | EndTr+TE80 N = 12 | |
| SOD [U/mg protein][a] | 9.45 ± 1.01 | 8.82 ± 0.45 | 8.02 ± 0.88[**] | $F_{2,28} = 8.26, p = 0.002$ |
| CAT [U/mg protein][a] | 8.85 ± 2.40 | 4.00 ± 1.35[***] | 2.99 ± 1.55[***] | $F_{2,28} = 30.9, p < 10^{-3}$ |
| GPx [U/mg protein][a] | 310 ± 66 | 289 ± 68 | 281 ± 65 | $F_{2,28} = 0.52, p = 0.60$ |
| GR [mU/mg protein][a] | 10.31 ± 3.14 | 9.22 ± 4.21 | 9.55 ± 4.26 | $F_{2,28} = 0.20, p = 0.82$ |
| GSH [mol/g tissue][b] | 4.24 ± 1.20 | 3.30 ± 0.48[*] | 3.77 ± 0.50 | $F_{2,28} = 3.37, p = 0.049$ |
| $\alpha$-Tocopherol [nmol/g tissue][b] | 53.2 ± 9.4 | 55.8 ± 5.3 | 50.7 ± 4.3 | $F_{2,28} = 1.54, p = 0.23$ |
| $\gamma$-Tocopherol [nmol/g tissue][b] | 0.99 ± 0.19 | 1.06 ± 0.22 | 0.96 ± 0.19 | $F_{2,28} = 0.76, p = 0.48$ |
| TBARs [nmol MDA/g tissue][b] | 839 ± 290 | 962 ± 56 | 1245 ± 325[**,‡] | $F_{2,28} = 6.91, p = 0.004$ |

Notes.

SOD, superoxide dismutase; CAT, catalase; GPx, glutathione peroxidase; GR, glutathione reductase; GSH, reduced glutathione; TBARs, thiobarbituric acid-reactive substances (lipid peroxidation products); EndTr, endurance training; TE8, 8 mg/kg$_{BW}$/week of intramuscular testosterone enanthate; TE80, 80 mg/kg$_{BW}$/week of intramuscular testosterone enanthate.

[a]per mg of protein in liver homogenate supernatant

[b]per g of liver wet weight

[*] $p < 0.05$, [**] $p < 0.01$, [***] $p < 0.001$ vs. the respective EndTr group value; ‡ $0.05 < p \leq 0.08$ vs. the respective EndTr+TE8 group value; Tukey's test.

### Tissue antioxidant enzymatic and nonenzymatic indices of liver status

TE treatment resulted in a slight drop in mean SOD activity and a marked lowering of mean CAT activity (Table 2). These activities also showed a significant negative correlation with the weekly TE dose across all three EndTr groups combined ($R_S = -0.61$ and $R_S = -0.78$, respectively; $p < 0.001$, $N = 31$ for both). In contrast, there was no sizable TE treatment-related difference in mean GPx and GR activities between these groups and no sizable tendency for a correlation with weekly TE dose across the EndTr study cohort (Table 2).

Both TE-treated groups showed somewhat lower liver GSH content than that in the TE-untreated group, which difference reached significance for the low-dose TE treatment. No tangible difference was found between the respective $\alpha$- or $\gamma$-tocopherol levels. As compared with hepatic TBARS content in the TE-untreated rats, that in the low-dose TE-treated rats was but nonsignificantly higher (+15%), while that in the other TE-treated group was significantly and much (+48%) higher (Table 2). Across all three EndTr groups combined, the TBARS content positively correlated with the TE dose received ($R_S = 0.51$, $p < 0.01$, $N = 31$).

### Blood serum enzymatic indices of liver status

There was no significant difference in serum AST or ALT activity between the various EndTr groups. Across all three EndTr groups combined, these activities did not or but poorly correlated with the weekly TE dose ($R_S = 0.20$, $p = 0.27$, and $R_S = 0.36$, $p = 0.046$, respectively, $N = 31$ for both). The rats given high-dose TE treatment showed significantly higher serum ALP and GGT activities than their TE-untreated counterparts (Table 3). These activities correlated well with the weekly TE dose across the three EndTr groups

**Table 3** Comparison of blood serum AST, ALT, ALP and GGT activities between adolescent male rats given 6-week endurance training without or with concurrent testosterone enanthate treatment.

| Blood serum enzyme | Rat group | | | ANOVA results |
|---|---|---|---|---|
| | EndTr $N = 10$ | EndTr+TE8 $N = 9$ | EndTr+TE80 $N = 12$ | |
| AST [U/l] | $179 \pm 47$ | $178 \pm 26$ | $189 \pm 23$ | $F_{2,28} = 0.39, p = 0.68$ |
| ALT [U/l] | $61 \pm 14$ | $70 \pm 12$ | $75 \pm 15$ | $F_{2,28} = 2.96, p = 0.068$ |
| ALP [U/l] | $157 \pm 30$ | $164 \pm 11$ | $180 \pm 14^{*}$ | $F_{2,28} = 4.00, p = 0.030$ |
| GGT [U/l] | $1.14 \pm 0.25$ | $1.42 \pm 0.37$ | $1.79 \pm 0.33^{***,‡}$ | $F_{2,28} = 11.24, p < 10^{-3}$ |

**Notes.**

AST, aspartate aminotransferase; ALT, alanine aminotransferase; ALP, alkaline phosphatase; GGT, $\gamma$-glutamyltransferase; EndTr, endurance training; TE8, 8 mg/kg_BW/week of intramuscular testosterone enanthate; TE80, 80 mg/kg_BW/week of intramuscular testosterone enanthate.

$*$ $p < 0.05$, $***$ $p < 0.001$ vs. the respective EndTr group value; ‡ $0.05 < p \leq 0.08$ vs. the respective EndTr+TE8 group value; Tukey's test.

**Table 4** Occurrence of above-normal activities of blood serum AST, ALT, ALP and GGT in adolescent male rats given 6-week endurance training without or with concurrent testosterone enanthate treatment.

| Enzyme activity | Surrogate reference range[a] | Occurrence of excessive enzyme activity | | |
|---|---|---|---|---|
| | | EndTr rats | EndTr+TE8 rats | EndTr+TE80 rats |
| AST [U/l] | 124-163 | 6/10 (60%)$^{§§}$ | 7/9 (78%)$^{§§§}$ | 10/12 (83%)$^{§§§}$ |
| ALT [U/l] | 42-71 | 2/10 (20%)$^{¶}$ | 4/9 (44%)$^{§§}$ | 7/12 (58%)$^{§§,*}$ |
| ALP [U/l] | 114-196 | 1/10 (10%) | 0/9 (0%) | 2/12 (17%)$^{¶}$ |
| GGT [U/l] | 0.57-1.39 | 2/10 (20%)$^{¶}$ | 3/9 (33%)$^{§}$ | 11/12 (92%)$^{§§§,***,##}$ |

**Notes.**

AST, aspartate aminotransferase; ALT, alanine aminotransferase; ALP, alkaline phosphatase; GGT, $\gamma$-glutamyltransferase; EndTr, endurance training; TE8, 8 mg/kg_BW/week of intramuscular testosterone enanthate; TE80, 80 mg/kg_BW/week of intramuscular testosterone enanthate.

[a]Based on data from age-matched TE-untreated sedentary male Wistar rats (UTr, $N = 11$) from the same breeding colony (Cmd:(WI)WU outbred stock) and kept under same conditions; taken from Sadowska-Krępa et al., 2017, with permission.

¶ $0.06 \leq p \leq 0.08$, § $p < 0.05$, §§ $p < 0.01$, §§§ $p < 0.001$ vs. the respective UTr group value (0/11, 0% for AST, ALT and ALP, and 0/10, 0% for GGT); $*$ $p < 0.05$, $***$ $p < 0.001$ vs. the respective EndTr group value; ## $p < 0.01$ vs. the respective EndTr+TE8 group value; the one-tailed $z$-test for two proportions.

($R_S = 0.61$, $p < 0.001$, and $R_S = 0.69$, $p < 0.001$, respectively, $N = 31$ for both). There was no correlation between any two of these indices within this cohort ($R_S \leq 0.30$, $p \geq 0.10$, $N = 31$).

The occurrence of above-normal AST activity was high in the TE-untreated EndTr rats and even higher in their TE-treated counterparts. The occurrence of excessive activities of ALT and GGT but not ALP tended to increase in the TE-untreated EndTr rats and was significantly elevated in the TE-treated rats TE (Table 4). However, even the maximum activities exceeded the upper limit of the respective reference range but moderately: AST - by 48%, ALT - by 37%, ALP - by 6%, and GGT - by 65%.

## Blood serum lipid profile

Across the entire cohort of endurance-trained rats, weekly TE dose showed significant negative correlation with serum HDL-Ch level and Tot-Ch/HDL-Ch ratio ($R_S = -0.77$,

**Table 5  Selected blood serum lipid levels and lipid ratios in adolescent male rats given 6-week EndTr without or with concurrent testosterone enanthate treatment and in their naive counterparts.**

| Blood serum lipid or lipid ratio | Rat group | | | | ANOVA results |
|---|---|---|---|---|---|
| | UTr $N = 11$ | EndTr $N = 10$ | EndTr+TE8 $N = 9$ | EndTr+TE80 $N = 12$ | |
| Tot-Ch [mg/dl] | $87 \pm 10$ | $79 \pm 7$ | $75 \pm 10$ | $71 \pm 10$ | $F_{3,38} = 6.19, p = 0.002$ |
| LDL-Ch [mg/dl] | $40 \pm 9$ | $33 \pm 7$ | $36 \pm 11$ | $36 \pm 8$ | $F_{3,38} = 1.32, p = 0.28$ |
| HDL-Ch [mg/dl] | $30 \pm 3$ | $30 \pm 3$ | $24 \pm 4$[***] | $21 \pm 3$[***] | $F_{3,38} = 23.4, p < 10^{-3}$ |
| Tot-Ch/HDL-Ch | $2.99 \pm 0.48$ | $2.62 \pm 0.32$ | $3.27 \pm 0.87$[¶] | $3.43 \pm 0.68$[*] | $F_{3,38} = 3.56, p = 0.023$ |
| LDL-Ch/HDL-Ch | $1.39 \pm 0.39$ | $1.09 \pm 0.26$ | $1.62 \pm 0.70$[¶] | $1.75 \pm 0.54$[*] | $F_{3,38} = 3.55, p = 0.023$ |
| TG [mg/dl] | $71 \pm 16$[*] | $50 \pm 14$ | $61 \pm 18$ | $42 \pm 15$ | $F_{3,38} = 7.31, p < 10^{-3}$ |
| TG/HDL-Ch | $2.42 \pm 0.61$[¶] | $1.67 \pm 0.48$ | $2.72 \pm 1.14$[*] | $2.01 \pm 0.75$ | $F_{3,38} = 3.56, p = 0.023$ |

Notes.

UTr, age-matched drug-naive untrained (sedentary) male Wistar rats from the same outbred stock (WI(WU)Cmd); EndTr, endurance training; TE8, 8 mg/kg$_{BW}$/week of intramuscular testosterone enanthate; TE80, 80 mg/kg$_{BW}$/week of intramuscular testosterone enanthate; Tot-Ch, total blood serum cholesterol; LDL-Ch, low-density lipoprotein cholesterol; HDL-Ch, high-density lipoprotein cholesterol; TG, triglycerides.

¶ $0.06 < p < 0.08$. * $p < 0.05$, *** $p < 0.001$ vs. the respective EndTr group value, Dunnett's test.

$p < 0.001$, and $R_S = -0.51$, $p < 0.01$, respectively, $N = 31$ for both), and a tendency for weak correlation with TG but not LDL-Ch level ($R_S = -0.31$, $p = 0.09$, and $R_S = 0.15$, $p = 0.41$, respectively, $N = 31$ for both). Mean values of all tested lipid indices except HDL-Ch tended to be lower in the TE-untreated EndTr rats than in their UTr counterparts, but the difference reached significance only for TG level and neared significance for the TG/HDL ratio (Table 5). Both TE-treated EndTr groups compared to their TE-untreated counterpart showed slightly and nonsignificantly lower mean Tot-Ch but not mean LDL-Ch level, and significantly lower mean HDL-Ch level. Mean Tot-Ch/HDL-Ch and LDL-Ch/HDL ratios were significantly higher in the EndTr+TE80 rats than those in their TE-untreated counterparts and showed a similar tendency in the EndTr+TE8 rats. The latter showed a nonsignificantly higher mean TG level than their TE-untreated counterparts, while there was a tendency for a reverse difference between them and the EndTr+TE80 rats. Due to no significant difference in mean HDL-Ch level between the TE-treated groups, the mean TG/HDL-Ch ratio was significantly higher (+63%) in the low-dose TE-treated, but not in the EndTr+TE80 rats (+20%) than in their TE-untreated counterparts (Table 5).

For lack of established reference ranges for blood lipids in Wistar rats, we estimated these ranges as the mean ± 2S.D. of the levels found in the previously mentioned additional group of matched TE-untreated UTr rats. Except for an abnormally low TG level and TG/HDL-Ch ratio in a single rat, no out-of-range value was found in the TE-untreated EndTr group. In contrast, the TE-treated EndTr rat groups, and mainly that given the higher TE dose, showed an increased occurrence of below-normal TG, Tot-Ch, and especially HDL-Ch levels, occasionally associated with abnormally high LDL-Ch/HDL-Ch ratio (Table 6).

## DISCUSSION

The main finding of this work is that testosterone supplementation enhanced hepatic stress and adversely altered liver pro-oxidant/antioxidant balance in EndTr adolescent male rats. These effects were evidenced by the increased occurrence of elevated blood GGT, AST and

**Table 6** Occurrence of abnormal blood serum TotCh, LDLCh, HDLCh and TG levels in adolescent male rats given 6-week endurance training without or with concurrent testosterone enanthate treatment.

| Serum lipid index | Surrogate reference range[a] | Occurrence of abnormal values | | |
|---|---|---|---|---|
| | | **EndTr rats** | **EndTr+TE8 rats** | **EndTr+TE80 rats** |
| LDL-Ch [mg/dl] | 22-59 | 0/10 (0%) | 0/9 (0%) | 0/12 (0%) |
| HDL-Ch [mg/dl][b] | 24-35 | 0/10 (0%) | 5/9 (56%)[§§,**] | 9/12 (75%)[§§§,***] |
| Tot-Ch [mg/dl][b] | 68-107 | 0/10 (0%) | 1/9 (11%) | 4/12 (33%)[§,*] |
| Tot-Ch/HDL-Ch[c] | 2.03-3.95 | 0/10 (0%) | 2/9 (22%)[§,†] | 3/12 (25%)[§,*] |
| LDL-Ch/HDL-Ch[c] | 0.61-2.17 | 0/10 (0%) | 2/9 (22%)[§,†] | 3/12 (25%)[§,*] |
| TG [mg/dl][b] | 39-103 | 1/10 (10%) | 0/9 (0%) | 7/12 (58%)[§§,**,##] |
| TG/HDL-Ch | 1.21-3.64 | 1/10 (10%) | 2/9 (22%)[§] | 1/12 (8%) |

**Notes.**
[a] Derived from the UTr group data shown in Table 5
[b] All abnormal values were below the lower limit of the respective reference range
[c] All abnormal values were above the upper limit of the respective reference range
EndTr, endurance training; TE8, 8 mg/kg$_{BW}$/week of intramuscular testosterone enanthate; TE80, 80 mg/kg$_{BW}$/week of intramuscular testosterone enanthate; Tot-Ch, total blood serum cholesterol; LDL-Ch, low-density lipoprotein cholesterol; HDL-Ch, high-density lipoprotein cholesterol; TG, triglycerides.
§ $p < 0.05$, §§ $p < 0.01$, §§§ $p < 0.001$ vs. the respective UTr group value (0/11, 0%); † $p = 0.058$. * $p < 0.05$, ** $p < 0.01$, *** $p < 0.001$ vs. the respective EndTr group value; ## $p < 0.01$ vs. the respective EndTr+TE8 group value; the one-tailed z-test for two proportions.

ALT activities, worsened blood lipid profile, reduced liver SOD and CAT activities and GSH content, and elevated liver TBARS content. The higher TE dose significantly lowered BW gain and LW as compared to those in TE-untreated EndTr rats, but caused no substantial shift in the LW/BW ratio ($-5\%$, $p = 0.66$) and thus no apparent hepatotoxicity in EndTr rats. Earlier, we found a significantly reduced LW/BW ratio in high-dose TE-treated adolescent UTr rats (*Sadowska-Krępa et al., 2017*), which may have been due to the start of TE treatment at a younger age.

The training was performed at an intensity that allows maximal fat oxidation (*Purdom et al., 2018*), and the mean BW gains were less in all three EndTr groups than in the respective UTr groups, see *Sadowska-Krępa et al. (2017)*. Notably, mean BW gain was identical for the TE-untreated and low-dose TE-treated groups in both the EndTr and the Utr rat cohort, indicating no interference of this treatment with food intake and somatic growth. The lowered BW gain in the EndTr+TE80 rats may also have resulted from reduced appetite and food intake. Such effects causing inadequate compensation of energy expenditure were commonly found in treadmill-trained male rats (*Harpur, 1980*; *Molano et al., 1999*; *Foletto et al., 2015*). However, reduced food intake was seen as well in long-term stanozolol-treated sedentary male rats (*Yu-Yahiro et al., 1989*). Another cause of reduced weight gain could be androgen-induced loss of adipose tissue (mostly visceral) and liver fat through various mechanisms (*Yu-Yahiro et al., 1989*; *De Pergola, 2000*; *Hoyos et al., 2012*).

Elevated serum AST, ALT, ALP, and GGT are well-known markers of AAS hepatotoxicity (*Urhausen, Torsten & Wilfried, 2003*; *Ozer et al., 2008*; *Singh, Bhat & Sharma, 2011*). However, serum ALT and AST may also come from injured skeletal muscles (*Ozer et al., 2008*). Exercise training alone can elevate serum ALT and AST, but not GGT, in both men (*Pettersson et al., 2008*; *Romagnoli et al., 2014*) and male rats (*Pey et al., 2003*; *Chang*

*et al., 2013*). We found no significant difference between our TE-untreated EndTr rats and matching UTr rats in any of the four markers but AST activity that was higher in the former ($p = 0.024$, Student's $t$-test); for the UTr rats data, see *Sadowska-Krępa et al. (2017)*. In the present study, raised serum AST, ALT, and GGT but not ALP activities were found in a sizable subset of the EndTr+TE8 rats and most of the EndTr+TE80 rats. However, the mean AST and ALT activities in the latter were only higher by 1/3 than those in the TE-untreated UTr rats (*Sadowska-Krępa et al., 2017*). Even smaller and nonsignificant were the relative increases in the mean AST (+6%) and ALT activity (+23%) in the EndTr+TE80 rats as compared to those in their TE-untreated counterparts, while the respective increase in the mean GGT activity was fairly robust (+57%). Our data on the effects of the lower TE dose are in line with those reported by others. Namely, no sizable increase in serum AST, ALT, ALP, or GGT was found in adolescent male rats given 12 weeks of moderate-intensity EndTr and five intragastric doses weekly of 2 mg/kg$_{BW}$ of fluoxymesterone, methylandrostanolone, or stanozolol during the last eight weeks (*Molano et al., 1999*; *Pey et al., 2003*). Notably, these AAS are considerably more hepatotoxic than testosterone (*Hartgens & Kuipers, 2004*; *Russmann, Kullak-Ublick & Grattagliano, 2009*; *Büttner & Thieme, 2010*).

We have earlier found significant increases in the mean serum level of AST, ALT, and ALP, but not GGT, in 5-week-old UTr rats given 6-week high-dose TE-treatment (*Sadowska-Krępa et al., 2017*). In the present study, 60% of the TE-untreated EndTr rats showed elevated serum AST levels indicating an exercise-related leakage of the enzyme from skeletal muscles. Unexpectedly, there was no significant TE-treatment-related increase in mean AST and ALT levels, while the high-dose TE treatment-related increase in mean ALP level was nearly identical to that in the UTr rats. However, contrary to the latter, the EndTr+TE80 rats showed a much higher mean GGT level than their TE-untreated counterparts. Serum ALT and ALP levels correlated positively with TE dosage in both the EndTr (see the Results section) and the UTr study cohort ($R_S = 0.70$, $p < 10^{-3}$, and $R_S = 0.50$, $p < 10^{-2}$, respectively, $N = 37$ for both, see Supplemental Files, UTr_cohort_ser_enz_vs_TE_dose.xlsx). In contrast, weekly TE dose positively correlated with GGT but not AST level in the EndTr rats (this study), while the reverse was right in the UTr rats ($R_S = 0.06$, $p = 0.72$, $N = 34$, and $R_S = 0.74$, $p < 10^{-3}$, $N = 37$, respectively, see Supplemental Files, UTr_cohort_ser_enz_vs_TE_dose.xlsx). This disparity may relate to the slightly different mean age of the two rat cohorts at the start of TE treatment, or a training-related change in resistance of the circulating enzymes' sources, or both.

Being fairly ubiquitous (*Bataller-Sifre, Guiral-Olivan & Bataller-Alberola, 2011*), GGT alone has insufficient specificity as a serum marker of liver injury. In humans, it is usually raised in liver pathologies involving cholestasis and jaundice, including those caused by AAS (*Urhausen, Torsten & Wilfried, 2003*; *Lumia & McGinnis, 2010*). In rats, it is supposedly a better cholestasis marker than serum ALP but less reliable than in other species (*Ozer et al., 2008*). We found elevated serum ALP activity in only two out of 12 EndTr+TE80 rats, but only one of them showed moderately elevated serum GGT activity. Notably, testosterone or its esters rarely cause adverse hepatobiliary effects, except possibly in aging (*Nucci et al., 2017*).

Cell membrane-bound GGT is a vital part of hepatic antioxidant lines that sustain cellular GSH and cysteine homeostasis (*Zhang, Forman & Choi, 2005*). Its link with serum GGT is not known. Serum GGT was proposed as an aggregate marker of organismal oxidative stress caused by various diseases and environmental chemicals (*Lee & Jacobs Jr, 2009*; *Chang et al., 2013*). The TE treatment-related differences in serum GGT and the occurrence of excessive serum GGT levels were much larger in the EndTr cohort than those found earlier (*Sadowska-Krępa et al., 2017*) in the UTr cohort. However, there was no significant association between the levels of serum GGT and serum ALP, ALT, or AST. Hence, a significant part of the rise in serum GGT activity might not be due to liver stress or damage. Instead, it may indicate increased oxidative stress occurring in multiple organs.

Mean liver TBARS content was only slightly higher in the TE-untreated EndTr rats than in their UTr counterparts (*Sadowska-Krępa et al., 2017*); this could be related to the constitutively high liver metabolic activity and the corresponding oxidative stress. However, the relative differences in liver TBARS content between the EndTr+TE8 and EndTr+TE80 rats and their TE-untreated counterparts were modest (+15 and +48%, respectively) compared to those in their sedentary counterparts (+56 and +78%, respectively, see *Sadowska-Krępa et al., 2017*). Hence, the absolute mean TBARS levels were nearly equal in the corresponding TE-treated UTr and EndTr rat groups. Though the present data confirm the pro-oxidant action of supraphysiological TE doses, they also suggest an attenuation of the added oxidative stress in the EndTr rats.

Decreased activities of liver CAT and SOD in high-dose TE-treated rats and negative correlations of these enzyme activities with weekly TE doses proved a harmful action of massive testosterone supplementation on liver antioxidant enzymes in adolescent EndTr rats. Even more significant relative declines, including those in GPx and GR activities, were found in the heart of EndTr male adolescent rats given testosterone propionate doses that produced much higher serum TT levels. Their associated relative increases in left heart ventricle TBARS content exceeded these in the liver. However, the maximum absolute mean TBARS contents found in the two organs were nearly identical to the respective maximum found in the soleus muscle and far above that in the extensor digitorum longus (*Sadowska-Krepa et al., 2013*). This similarity may be related to the fact that the soleus muscle, the myocardium, and the liver, but not the extensor digitorum longus, rely almost entirely on oxidative metabolism in exercise.

While we found decreased hepatic GSH content in our TE-treated EndTr rats, stanozolol treatment was reported to increase hepatic GSH content in endurance-trained rats (*Pey et al., 2003*). The cause of the difference may be distinct pharmacological profiles of stanozolol and testosterone (*Fernández et al., 1994*; *Hartgens & Kuipers, 2004*; *Russmann, Kullak-Ublick & Grattagliano, 2009*; *Büttner & Thieme, 2010*), but also younger age and thus different reactivity of our rats at the beginning of androgen treatment. The present data contrast also with those from adolescent UTr rats, in which the same TE doses significantly elevated liver SOD activity (*Sadowska-Krępa et al., 2017*). It is likely that long-term TE treatment of adolescent male EndTr rats perturbed their development and/or boosted oxidative stress beyond the capacity of hepatic defenses. Of note, mean liver activities of SOD, CAT and GPx were higher ($p \leq 0.035$, one-tailed $t$-test) in the

TE-untreated EndTr rats than in their UTr counterparts (*Sadowska-Krępa et al., 2017*), and a similar tendency was found for GR ($p = 0.093$, one-tailed $t$-test). These findings implied that the training alone boosted the antioxidant defenses. A similar effect on liver SOD but not CAT and GPx activities was found in male rats of similar age, which were given 16-week EndTr of similar intensity (*Song, Igawa & Horii, 1996*). In our study, the positive effect of the training on liver SOD and CAT, but not GR and GPx activities, was dose-dependently reduced by concurrent TE treatment. This decline is consistent with enhanced liver oxidative stress in adolescent UTr rats (*Sadowska-Krępa et al., 2017*) and prompt deactivation of CAT and SOD by oxygen free radicals (*Salo et al., 1990*; *Escobar, Rubio & Lassi, 1996*).

GSH is the most abundant antioxidant and the key scavenger of reactive oxygen and nitrogen species in the liver (*Russmann, Kullak-Ublick & Grattagliano, 2009*; *Li et al., 2015*). In our studies, its content did not differ between the TE-untreated EndTr and UTr rats. However, in contrast to what we found in the UTr rats study (*Sadowska-Krępa et al., 2017*), TE treatment suppressed hepatic GSH content in EndTr rats, suggesting increased GSH use. GSH is also vital for sustaining reduced forms of some exogenous antioxidants, including the essential lipophilic antioxidant vitamin E (*Scholz et al., 1989*). Neither hepatic $\alpha$-tocopherol nor $\gamma$-tocopherol pool was significantly affected by TE treatment in EndTr rats, showing the efficacy of GSH in maintaining appropriate levels of these protectants against free-radical mediated liver damage (*Leo, Rosman & Lieber, 1993*). Interestingly, liver TBARS content negatively correlated with hepatic $\alpha$-tocopherol ($R_S = -0.48$, $p < 0.01$, $N = 31$) but not $\gamma$-tocopherol content ($R_S = -0.30$, $p = 0.11$, $N = 31$) within the entire EndTr cohort. This link may reflect the use of $\alpha$-tocopherol for scavenging reactive oxygen species formed, e.g., due to $\beta$-oxidation of fatty acids, and hence for breaking chain propagation and maintaining an adequate redox balance (*Niki, 2014*).

The EndTr-related shifts in blood lipid profile found in this study are usually linked to reduced atherogenicity in both humans (*Shephard & Johnson, 2015*; *Ruegsegger & Booth, 2018*) and rats (*Burneiko et al., 2006*; *Kazeminasab et al., 2017*). The changes in blood lipid profile caused by concurrent TE treatment evidenced a reversal of the beneficial effects of the training and worsening of some training-unaffected characteristics except for the tendency of TG level and TG/HDL ratio to drop with the high TE dosage. The latter was likely the result of long-term action of supraphysiological serum TT level on male adipose tissue metabolism and the ensuing decreased body and liver fat contents and availability.

## CONCLUSIONS

The present results show that long-term systemic high-dose testosterone treatment harms liver antioxidant defense systems and function in adolescent male rats undergoing endurance training. Namely, it abolishes or markedly attenuates most studied metabolic benefits from the training, causing a negative shift in liver pro-oxidative/antioxidative balance evidenced by reduced SOD and CAT activities, raised hepatic TBARS level, and elevated serum GGT activity. These changes suggest increased oxidative stress that likely occurs in other organs as well and may increase morbidity later in life. The same may occur

in human male adolescents using massive testosterone or other AAS supplementation as a shortcut to improved sports performance and/or a more muscular physical appearance.

### Funding
This study was supported by statutory funds from the Jerzy Kukuczka Academy of Physical Education, Katowice, Poland. The funders had no role in study design, data collection and analysis, decision to publish, or preparation of the manuscript.

### Grant Disclosures
The following grant information was disclosed by the authors:
Jerzy Kukuczka Academy of Physical Education, Katowice, Poland.

### Competing Interests
The authors declare there are no competing interests.

### Author Contributions
- Ewa Sadowska-Krępa and Barbara Kłapcińska conceived and designed the experiments, analyzed the data, authored or reviewed drafts of the paper, and approved the final draft.
- Anna Nowara, Sławomir Jagsz, Izabela Szołtysek-Bołdys and Małgorzata Chalimoniuk performed the experiments, authored or reviewed drafts of the paper, and approved the final draft.
- Józef Langfort analyzed the data, authored or reviewed drafts of the paper, and approved the final draft.
- Stanisław J Chrapusta analyzed the data, prepared figures and/or tables, authored or reviewed drafts of the paper, and approved the final draft.

### Animal Ethics
The following information was supplied relating to ethical approvals (i.e., approving body and any reference numbers):

The study protocol has been fully accepted by the IV Local Ethics Committee for Animal Experimentation in Warsaw under Permit No. 38/2011 of 13 June, 2011.

### Data Availability
The raw data are available in a Supplemental File.

### Supplemental Information
Supplemental information for this article can be found online at http://dx.doi.org/10.7717/peerj.10228#supplemental-information.

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
