# Peer review of "High-dose testosterone supplementation disturbs liver pro-oxidant/antioxidant balance and function in adolescent male Wistar rats undergoing moderate-intensity endurance training"

_PeerJ, doi:10.7717/peerj.10228_

## Round 0.1 · original submission · Minor Revisions

Based on the advice received, I have decided that your manuscript could be reconsidered for publication should you be prepared to incorporate minor revisions.

Reviewer 1 ·

Basic reporting

1. The manuscript is clearly written and is comprehensible. The introduction is sufficient to orient towards the aims and objectives of the work.

Experimental design

1. The research question is well-stated and the methods are sufficiently elaborated.
2. Proper statistics have been applied.

Validity of the findings

1. Is it known what could be a possible trigger for oxidative stress in liver after high testosterone supplementation?
2. Given the known role of Androgen receptor signaling in regulating lipid and glucose homeostasis through liver cells, the authors should discuss the effects of supraphysiological levels of testosterone in the light of their results. What about the insulin sensitivity at high testosterone supplementation? It may also pose detrimental effects.
3. Testosterone can have effects on multiple signaling pathways and organs, but since this paper is more focused on oxidative stress and liver injury, authors should check how Nrf2 signaling (which is a master regulator of antioxidant defenses) is affected. We see that high dose TE leads to increased lipid oxidation (TBARS) together with decrease in antioxidant defenses like CAT, it would be interesting to know if this is due to down-modulation of cellular defenses or exhaustion of resources.

Additional comments

1. The discussion is too elaborate, cutting it short wherever possible would make the manuscript more coherent.
2. Line 211 replace ‘that in’ with than.

·

Basic reporting

The English language used in the manuscript is clear, mainly in the discussion.

Your introduction is very good, however it is necessary to insert the article reference at lines 62-63 and 115-116 and remove website. I also suggest that you improve the description at lines 62-63 and 115-116 to provide trustworthiness for your study (specifically, you should insert the websites in supplementary data)

Experimental design

If the testosterone dilution at line 135 are included in another study, please cite him in this section.

Considering the methods section, experimental design, a suggestion is to group information from line 159-161 with line 146 to the information not to get lost. In the same section, the authors have data of food or water consumption? If do not, a suggestion to the authors is to use in next studies the control of water and food consumption weekly or, if it is possible, daily. This data control can help to better understand the BW of groups EndTr, mainly with the use of testosterone protocol. For observation, another suggestion to next studies is to change bedding three times a week.

Validity of the findings

The manuscript “High-dose testosterone supplementation disturbs liver pro-oxidant/antioxidant balance and function in adolescent male Wistar rats undergoing moderate intensity endurance training (#46526)” presents a very important data about the use of testosterone in the adolescent phase. This data can contribute to evidence the precautions about the use of testosterone.

Additional comments

To value your paper, another suggestion is to create a figure that explains your results and summarizes the discussion theory, if it is possible, insert in the article or supplementary data.

---

## Round 0.2 · accepted · Accept

Based on the advice received, I have decided that your manuscript could be accepted for publication in its present form.